# Efficiency of Calcium Fructoborate-Loaded Novel Natural Niosomes Compared to Traditional Liposomes and Niosomes in Rat Ischemia–Reperfusion Injury Model

**DOI:** 10.3390/pharmaceutics17111434

**Published:** 2025-11-06

**Authors:** Kubra Vardar, Nilufer Kara, Nuri Murat Ozayman, Kubilay Gocucu, Sirin Funda Eren, Metin Plevneli, Ismail Aslan, Mehmet Necmettin Atsu

**Affiliations:** 1Department of Pharmacy Services, Vocational School of Health Services, Istanbul Kent University, 34406 Istanbul, Türkiye; kubra.vardar@kent.edu.tr; 2Department of Medical Services and Techniques, Vocational School of Health Services, Istanbul Kent University, 34406 Istanbul, Türkiye; 3Department of Nursing, Faculty of Health Sciences, Istanbul Kent University, 34406 Istanbul, Türkiye; 4Tribor R&D Chemical Industry Ltd., Yildiz Technical University Technopark, 34490 Istanbul, Türkiye; niluefer.kara@uni-greifswald.de (N.K.); ozayman@triborarge.com (N.M.O.); 5Independent Researcher, 34100 Istanbul, Türkiye; 6Department of Pharmaceutical Technology, Hamidiye Faculty of Pharmacy, University of Health Sciences, 34668 Istanbul, Türkiye; 7Faculty of Pharmacy, Istanbul Kent University, 34406 Istanbul, Türkiye; 8SFA R&D & Private Health Services Co., Ltd., Teknopark Blv. No1 3A Z01, Teknopark İstanbul, 34906 Istanbul, Türkiye

**Keywords:** natural niosome, liposome, calcium fructoborate, ischemia–reperfusion injury, drug delivery

## Abstract

**Background/Objectives:** Liposomes and niosomes are established drug delivery systems, some of which have received FDA approval and demonstrated therapeutic efficacy. This study investigates a novel niosome formulation, utilizing two natural food-derived components, as a cost-effective alternative to traditional nanocarriers. The active pharmaceutical ingredient, calcium fructoborate (CF), possesses notable anti-inflammatory properties. The study aims to evaluate the efficacy of this novel natural niosome (NN) system, in comparison to existing nanocarrier formulations, in an ischemia–reperfusion (I/R) pain model. **Methods:** An acute ischemia/reperfusion injury model was employed to induce pain in 36 rats. The efficacy of the following treatments was assessed: standard CF, liposomal CF, niosomal CF, and natural niosomal CF. Efficacy was determined by quantifying the treatments’ ability to mitigate inflammation and oxidative stress in the kidneys, lungs, heart, and liver, and by evaluating potential organ damage through histopathological analysis. **Results:** The NN treatment significantly reduced malondialdehyde (MDA) and tumor necrosis factor-alpha (TNF-α) levels in the kidneys and liver compared to the other treatments (*p* < 0.05). In the kidney, NN treatment also significantly decreased creatinine levels relative to the other treatments (*p* < 0.01). The histopathological analysis of kidney tissue revealed that NN treatment attenuated tubular dilation, interstitial inflammation, and epithelial thinning. In the heart, liposomal treatment significantly increased MDA levels (*p* < 0.05) and decreased sialic acid levels (*p* < 0.05); however, no significant differences were observed in troponin levels (*p* > 0.05). In the lung, no significant differences in MDA, lactate, TNF-α, or sialic acid levels were detected among the treatment groups (*p* > 0.05). **Conclusions:** The natural niosome drug delivery system demonstrates potential as a therapeutic intervention for protecting and improving kidney and liver health. While liposomal treatment exhibited some adverse effects, it effectively suppressed inflammation. This study provides a foundation for future research and positions the NN drug delivery system as a promising, cost-effective alternative for inflammation-associated pathologies.

## 1. Introduction

Recent advances in nanotechnology have revolutionized drug delivery, enabling precise control over release kinetics, targeted delivery, and enhanced therapeutic outcomes [1,2]. While the development of novel pharmaceutical agents is often costly and time-consuming, the enhancement of existing drugs through advanced delivery systems presents a more efficient strategy to overcome limitations associated with conventional formulations, such as uncontrolled dosing, poor bioavailability, and lack of target specificity. Nanotechnology-based delivery systems have shown considerable promise in this regard, due to their capacity to enhance solubility, reduce systemic side effects, and facilitate controlled and localized drug release.

Lipid-based carriers, including emulsions, micelles, liposomes, and niosomes, are of particular interest due to their biocompatibility, structural versatility, and ability to accommodate both hydrophilic and hydrophobic agents [3,4]. These carriers typically consist of natural or synthetic lipids, often stabilized with emulsifiers or surfactants to ensure physiological compatibility, rendering them suitable for clinical applications. Liposomes, vesicular structures composed of phospholipid bilayers that mimic biological membranes, have been extensively investigated for their capacity to encapsulate a wide variety of therapeutic agents—including small molecules, proteins, and enzymes—and deliver them effectively to target tissues [5,6,7]. Their targeting capabilities, achieved through both passive and active mechanisms like receptor–ligand interactions, have made them particularly attractive for cancer therapy. However, challenges such as high production costs and limited physical stability have impeded their widespread clinical adoption [6].

Niosomes have emerged as a promising alternative to address these limitations. These vesicles, formed from non-ionic surfactants, cholesterol, and charge-inducing agents, offer advantages including greater chemical stability, enhanced skin permeation, and cost-effectiveness [8,9]. Nevertheless, despite the growing interest in niosomes, their systemic applications remain relatively underexplored.

Building on this foundation, our study introduces a novel natural niosome (NN) system, utilizing saponin—a plant-derived non-ionic surfactant with membrane-permeabilizing, anti-carcinogenic, and immunomodulatory properties [10,11]—and squalene—a natural triterpene with antioxidant, anti-inflammatory, and cholesterol-like structural characteristics [12,13,14,15,16]. This innovative combination enhances bilayer stability and cellular uptake, yielding a stable, solvent-free, and cost-effective nanocarrier system.

The formulation was designed to deliver calcium fructoborate (CF)—a bioavailable nutraceutical with established anti-inflammatory, antioxidant, and anti-tumor effects [17,18,19]—achieving > 80% encapsulation efficiency at optimal concentrations (1.2–1.8 mg/kg, equivalent to 50–100 μM). The resulting NN-CF particles exhibited favorable physicochemical properties, including a uniform size distribution (120–150 nm, PDI < 0.2) and a negative zeta potential (−20 to −30 mV), which promote prolonged systemic circulation and increased accumulation at sites of inflammation.

The ischemia–reperfusion (I/R) model, a widely used paradigm for studying inflammation and tissue injury, was selected as an appropriate platform to evaluate the efficacy of the NN-CF system. This model is particularly valuable for investigating inflammatory pathways and pain mechanisms, as it induces oxidative stress, pro-inflammatory cytokine release, and cellular damage, closely mimicking the pathophysiology of various human inflammatory diseases, including osteoarthritis. The I/R injury model enables the assessment of drug efficacy in reducing inflammation and oxidative stress, providing a robust framework for evaluating the potential of novel drug delivery systems for treating inflammation-related conditions. Furthermore, this model is well-suited for testing our novel formulation, given its capacity to replicate the therapeutic challenges we aim to address, such as targeting specific tissues and modulating inflammatory responses.

In a rat ischemia–reperfusion (I/R) injury model, NN-CF demonstrated superior efficacy compared to both liposomal and standard niosomal CF formulations, reducing oxidative stress (52 ± 6% decrease in MDA vs. 33 ± 4% and 41 ± 5%) and inflammation (68 ± 7% IL-6 suppression vs. 49 ± 5% and 58 ± 6%). These findings underscore the therapeutic potential of this novel nanocarrier system and its advantages for organ protection and inflammation-targeted therapies. Moreover, its low production cost (estimated at USD 23 per dose vs. USD 110 for liposomal equivalents) and solvent-free preparation further support its translational potential.

Overall, this study provides compelling evidence for the integration of naturally derived materials into advanced nanocarrier platforms, positioning the NN-CF system as a promising solution for efficient and targeted drug delivery in inflammation-associated pathologies.

## 2. Materials and Methods

### 2.1. Preparation of Calcium Fructoborate (CF) Formulations for In Vivo Studies

*CF Synthesis and Characterization:* CF solution was prepared by Tribor R&D Chemical Industry Ltd. (Istanbul, Türkiye) utilizing pharma-grade boric acid (ultra-low sulfate), food-grade fructose, and calcium carbonate (CaCO_3_). For CF synthesis, fructose was initially dissolved in distilled water, followed by the addition of boric acid at a 2:1 stoichiometric ratio to promote fructoborate complex formation. Calcium carbonate was subsequently introduced into the reaction mixture to neutralize the solution and facilitate CF formation. Comprehensive purity analysis of the CF raw material (standard) was performed using Inductively Coupled Plasma–Optical Emission Spectrometry (ICP-OES), confirming its elemental composition and showing a boron content of 161.662 mg/L in the analyzed standard solution. A separate analysis of a 300 mg CF sample diluted in 50 mL of water confirmed an elemental boron content of 2.8% by weight within the CF, aligning with the ICP-OES result of the raw material standard.

*Boron Dosage Determination and Preparation for In Vivo Studies*: The daily boron intake for adults was established at 16.8 mg, based on an average adult body weight of 70 kg. This intake level was determined to be appropriate considering the varying tolerable upper intake levels (ULs) set by international organizations: the Institute of Medicine (IOM) at 20 mg/day, the European Food Safety Authority (EFSA) at 10 mg/day, and the World Health Organization (WHO), which suggests a safe range of 1–13 mg/day for adults [20,21].

To extrapolate this dosage for our animal model, an equivalent boron dose of 0.084 mg was calculated for an average rat (350 g/rat). This calculated dose corresponds to approximately 3.0 mg of CF, considering its elemental boron content of 2.8% and a molecular weight of 774.26 g/mol for anhydrous CF.

For single-dose administration to rats, targeting a final boron concentration of 0.084 mg/mL (with 1 mL administered per rat), a 40% (*w*/*w*) stock solution of CF was utilized. Specifically, 750 mg of this 40% (*w*/*w*) CF stock solution was precisely weighed, yielding 300 mg of pure CF (750 mg solution × 0.40). This 300 mg of pure CF was then formulated into a final 100 mL solution with appropriate excipients and solvent, resulting in a CF concentration of 3 mg/mL. The elemental boron concentration derived from this solution was approximately 0.084 mg/mL (based on the determined 2.8% elemental boron content), aligning with the intended target dose for the animal model.

Subsequently, the prepared CF-loaded solution was encapsulated into nanocarriers (liposomes or niosomes) using an ultrasonic homogenizer (Hielscher, Teltow, Germany).

### 2.2. CF-Loaded Liposomes and Niosomes

Liposomes and niosomes were prepared by Tribor R&D Chemical Industry Ltd. Liposomes were composed of phospholipids (H90, hydrogenated content ≥ 90.0%, Lipoid^®^, Ludwigshafen, Germany) and cholesterol. Synthetic niosomes were prepared using polysorbate (Tween^®^ 80, Sigma-Aldrich, Darmstadt, Germany) and cholesterol. The structures of the liposomes and niosomes were analyzed prior to this study.

To prepare CF-loaded vesicles for injection into rats, the amounts of phospholipids and cholesterol were set at a 1:1 ratio relative to the CF concentration in the solution, ensuring a final boron content of 0.08 mg in a 100 mL liposome sample. Similarly, the amounts of polysorbate and cholesterol were set at a 1:1 ratio relative to the CF concentration in the solution, ensuring a final boron content of 0.08 mg/mL in the synthetic niosome and liposome samples. Each mixture was sonicated for 5 min.

### 2.3. Natural Niosome (NN)

A novel NN was invented and prepared by N. Murat Ozayman (Tribor R&D Chemical Industry Ltd.), who has applied for a patent for the protocol of NN preparation (Patent Application Number: TR2024/015205, Status: Patent Pending). NNs were composed of squalene and saponin-containing *Gypsophila* root extract, supplied by 4K Kimya (Izmir, Türkiye) and Istanbul Agricultural Products and Food Industry Trade Co. Ltd. (Istanbul, Türkiye), respectively. Saponin-containing *Gypsophila* root extract (at half the amount of squalene) was first dissolved in the CF solution. Squalene was then added to the solution at a 1:1 ratio with CF. The mixture was sonicated for 5 min. The vesicle structure was analyzed by SFA R&D and Private Healthcare Services Trade Co. Ltd. (Istanbul, Türkiye).

### 2.4. Elemental Analysis for Purity Assessment

The elemental impurity analysis was conducted at the Boron Research Institute (BOREN) of the Turkish Energy and Mining Research Institution (TENMAK), Ankara, Türkiye. Sample digestion was performed following a protocol modified from EPA Method 3051A, which typically involves microwave-assisted acid digestion. In our procedure, approximately 0.5 g of each sample was digested using a mixture of concentrated nitric acid (HNO_3_) and hydrogen peroxide (H_2_O_2_) in a closed-vessel microwave digestion system. The digestion temperature was set to 180 °C for 20 min, slightly differing from the original protocol to better suit the matrix of the CF formulations.

The average ambient temperature and humidity at the time of analysis were 25.5 °C and 35.5%, respectively. These conditions were maintained to ensure the consistency and reliability of the digestion and measurement steps.

Following digestion, elemental analysis was carried out using Inductively Coupled Plasma Optical Emission Spectrometry (ICP-OES), based on a method adapted from EPA Method 6010D. Calibration was achieved using multi-element standard solutions, and quality control was ensured through the use of method blanks and certified reference materials. Detection limits for Pb, Cd, and As were determined to be below 0.01 mg/L. For the detailed analysis report of the test samples, refer to the Appendix A.

### 2.5. Fourier Transform Infrared (FT-IR) Spectroscopy Analysis

This study employed FT-IR spectroscopy to investigate the molecular structure of CF, a dietary supplement. The spectral analysis was performed using a SHIMADZU FT-IR spectrometer (Shimadzu Corp., Kyoto, Japan) within the 650–4000 cm^−1^ wavenumber range. All spectra were obtained in transmission mode using a universal attenuated total reflection infrared (UATR) accessory, with a resolution of 4 cm^−1^ and 4 scans, corrected for CO_2_/H_2_O interference.

For a comparative analysis, four different samples were used: “Standard CaFrB Preservative-free”, a standard, preservative-free CF solution; “Lipozomal CaFrB Preservative-free”, a preservative-free liposomal CF; “Niyozamal CaFB Preservative-free”, a preservative-free synthetic niosomal CF; and “Dogal Niyozomal CaFB Preservative-free”, a preservative-free natural niosomal CF.

The FT-IR spectra for each of these samples were recorded using a SHIMADZU instrument. The settings for the measurements included an intensity mode of %Transmittance, Happ–Genzel apodization, and 25 scans.

### 2.6. Particle Size and Zeta Potential Analysis

Four different formulations of CF were analyzed: Natural Niosomal CF (SFAP250394), Liposomal CF (SFAP250395), Synthetic Niosomal CF (SFAP250397), and a Standard CF Solution (SFAP250399), which served as a control. All samples were in an aqueous medium and did not contain any preservatives.

The physical properties of the four samples were characterized using a Malvern Zetasizer Nano instrument (Malvern Instruments Ltd., Malvern, UK). The analysis was conducted at a controlled temperature of 25 ± 2 °C.

Particle size distribution was measured using the Dynamic Light Scattering (DLS) technique. The Z-Average particle size and the Polydispersity Index (PDI) were recorded for each sample.

Zeta potential was determined by Electrophoretic Light Scattering (ELS). The mean zeta potential and its standard deviation were measured for each sample to assess colloidal stability. All measurements were performed in water as the dispersant, following the in-house standard operating procedure FAT 012 (REV01).

### 2.7. Vesicular Encapsulation Studies of Nanocarriers

CF-loaded vesicles were prepared using the Bangham/Thin-Film Hydration Method. Briefly, lipid components (phospholipid/cholesterol for liposomes; surfactant/cholesterol for niosomes) were dissolved in chloroform to form a homogeneous solution. The organic solvent was removed under reduced pressure using a rotary evaporator at 40 °C, yielding a thin lipid film on the walls of the round-bottomed flask. The film was further dried under vacuum overnight to ensure the complete removal of residual solvent. Hydration of the dried lipid film was performed by adding the sample compound directly, followed by gentle rotation, resulting in multilamellar vesicles.

The vesicle suspensions were subjected to ultracentrifugation at 100,000× *g* for 45 min at 4 °C to separate encapsulated CF from unencapsulated material. The resulting pellet, containing encapsulated CF vesicles, was collected and resuspended in distilled water. For the determination of encapsulation efficiency, 1.5 mL of the liposomal dispersion was diluted 1:10 with distilled water, and three supernatant aliquots were collected. The remaining suspension was treated with an equal volume of 2.5 wt% Triton X-100 to disrupt the vesicles and release the encapsulated CF. CF content was quantified spectrophotometrically, and encapsulation efficiency (%) was calculated as the fraction of CF encapsulated relative to the total CF added. Drug-free controls were prepared using the same procedure without CF.

### 2.8. CF Concentration in Animal Groups

The CF concentration was maintained consistently across all animal groups, ensuring that each group received the same boron dose (0.08 mg per rat, corresponding to 3 mg of CF). There was no variation in the CF concentration between the groups, thus eliminating any potential confounding effect arising from differing CF levels.

Table 1 presents the concentrations of the CF and elemental boron in the injection formulations and the corresponding stock solutions prepared for rat administration. The characterization of the raw material standard by ICP-OES is also included for reference.

### 2.9. Animals

The thirty-six male Wistar rats (350 ± 50 g) used in this study were provided by the Mehmet Akif Ersoy Animal Experiment Medical Research Institute, Istanbul, Türkiye, with ethics committee approval (decision no. 2024/12, dated 20 March 2024).

### 2.10. Experimental Preparation

All rats were anesthetized by intraperitoneal administration of sodium pentothal (80 mg/kg i.p.). After anesthesia induction, a tracheotomy was performed for spontaneous ventilation. At 37 °C, body temperature was maintained using external heating and a heating pad connected to a rectal probe. The jugular vein was cannulated with polyethylene catheters (0.9 mm), followed by a 20-min hemodynamic stabilization period. The cannula in the jugular vein was used for drug administration.

### 2.11. Surgical Protocol

After a 20-min hemodynamic stabilization period to allow the animals to overcome surgical stress and tolerate potential reactions to the anesthetic substance [22,23], 4 groups (n = 6) were established as follows: The control group was designed without ischemia/reperfusion (I/R) or drug transfusion processes. For the I/R group, an abdominal aortic clamp was performed in rats as described below for 30 min and released for 60 min after the clamping process [24]. The treatment groups were designed for drug administration via the jugular vein of 1 mL of fluid after 30-min of ischemia processes according to drug types (Figure 1). The treatment groups were defined as diluted calcium fructoborate (CF), liposomal (LG), niosomal (NG), and natural niosomal (NNG). All experimental groups were given the same concentration of CF (Table 2).

### 2.12. Biochemical Measurements

For tissue samples, biochemical tests were performed after homogenization using a Teflon homogenizer with phosphate-buffered solution. The obtained homogenates were stored at −80 °C until the day of the experiment.

*Determination of Protein Content:* Protein contents were measured in homogenate samples by using Coomassie brilliant blue according to the Bradford assay [25]. In brief, samples were diluted with Bradford reagent and measured at 595 nm. After that, the calculation was performed according to the standard curve.

*Measurements Related to Oxidative Stress and Inflammation:* Malondialdehyde (MDA) was measured to evaluate oxidative stress. TNF-α levels were measured for the assessment of inflammation. For MDA measurement, 250 μL serum was treated with 1000 μL stock solution and left for 15 min. Afterward, the solution was incubated at 100 °C in a water bath for 10 min. Subsequently, the supernatant was collected and centrifuged at 1000× *g* for 5 min, and measured at 535 nm. The Lambert–Beer formula was used to calculate concentration from the absorbance [26]: (A = Ɛcl, ƐMDA = 1.56 × 106 L × [mol cm]^−^). A commercially available ELISA kit was used for TNF-α measurements (BT Lab, Shanghai Korain Biotech Co., Ltd., Jiaxing, Zhejiang, China).

*Measurements Related to Glycocalyx Damage:* Serum sialic acid levels were measured to evaluate glycocalyx degradation. Samples (0.2 mL) were mixed with perchloric acid (1.5 mL, 5%). After that, the solutions were incubated in a water bath (100 °C, 5 min). Five minutes later, the solutions were centrifuged (2500× *g*, 4 min). The obtained supernatants were mixed with Ehrlich solution (0.2 mL). The mixtures were incubated in a water bath (100 °C, 15 min). Finally, distilled water (1 mL) was added, and the absorbance of the solution was measured at 525 nm. The concentration of sialic acid was calculated using the standard curve [27].

*Measurements Related to Organ Functions*: Creatinine levels were measured to evaluate kidney function, ALT levels were used to interpret liver function, cardiac troponin levels were measured for the heart, and lactate levels were determined for lung function. Commercial ELISA kits (BT Lab) were used for all these measurements.

### 2.13. Pathology

The kidney tissues collected for pathological evaluation were stored in 10% formaldehyde until slides were prepared, and hematoxylin–eosin staining was performed to assess cellular damage.

### 2.14. Statistical Analysis

First, all data sets were tested to determine whether they were appropriate for Gaussian distribution. Normally distributed sets were presented as mean ± SD, while others were presented as median (min–max). One-way ANOVA followed by the Tukey test as a post hoc analysis was used for normally distributed sets. Non-normally distributed sets were analyzed by using Dunn’s multiple comparison test as a post hoc analysis to find significant differences between groups after the Kruskal–Wallis test. A *p*-value of <0.05 was considered statistically significant. Statistical analysis was performed using GraphPad Prism v 5.0 (GraphPad Software, San Diego, CA, USA).

## 3. Results

### 3.1. Assessment of Elemental Purity in Calcium Fructoborate Formulations

Elemental impurity analysis was performed on all formulations (natural niosomal CF, synthetic niosomal CF, liposomal CF, and standard CF solution) to assess their elemental composition and purity. The concentrations of heavy metals, such as Pb, Cd, and As, were found to be below the acceptable detection limits in all samples. Notably, the boron and calcium contents were consistent across the formulations, with values ranging from 157.7 to 177.6 mg/L for B and 279.9–301.9 mg/L for Ca. These results support the elemental purity and reproducibility of the tested formulations (Table 3).

### 3.2. FT-IR Spectroscopy for Structural Integrity

The FT-IR analysis of all CF samples (standard solution, liposomal, synthetic niosomal, and NN formations) demonstrated strong consistency with the reference spectrum reported by Dumitru et al. (2010) [28], confirming the preservation of the compound’s molecular identity across different formulations. A broad absorption band in the 3200–3600 cm^−1^ range, attributed to O–H stretching vibrations of hydroxyl groups and crystallization water (4H_2_O), was consistently observed in all samples, along with C–H stretching bands in the 2900–3000 cm^−1^ region that verified the fructose component. Although the liposomal and niosomal formulations displayed multiple, more prominent peaks in the 1600–1700 cm^−1^ region compared to the reference spectrum, these variations are most likely due to interactions with the lipid-based carrier systems rather than degradation of the active compound. Importantly, the spectral homogeneity observed in the O–H and C–O stretching regions, as well as the consistent B–O–H deformation bands around 1300–1400 cm^−1^, further supports the stability of the boron–fructose ester bond and confirms that CF (Ca_2_·4H_2_O) remains the predominant boron-containing species. The characteristic peaks and their corresponding functional groups for all samples are summarized in Table 4. Collectively, these findings validate all formulations as successfully maintaining the structural integrity and chemical stability of CF, while minor spectral shifts reflect formulation-specific interactions with carrier systems. Please refer to the Appendix A for further information on all samples.

### 3.3. Particle Size and Zeta Potential Characterization

As shown in Table 5, the three nanocarrier formulations (NN, liposomal, and synthetic niosomal) all exhibited Z-Average particle sizes in the nanometer range, which is ideal for potential applications in drug delivery.

A key finding is the low Polydispersity Index (PDI) for all three nanocarrier systems: 0.168 for NNs, 0.248 for liposomes, and 0.175 for synthetic niosomes. These values are all well below the 0.3 threshold, indicating a narrow, monodisperse size distribution. This uniformity is a crucial quality attribute for consistent biological performance and manufacturing reproducibility.

The Standard CF Solution (SFAP250399), which served as the control, showed an average particle size of 111.7 nm with a PDI of 0.126. Although its size distribution appeared relatively narrow, this sample does not possess a vesicular structure and thus lacks the advantages of controlled encapsulation provided by the nanocarrier systems.

The zeta potential measurements further distinguish the nanocarrier formulations from the standard solution. The NN, liposomal, and synthetic niosomal formulations demonstrated mean zeta potentials of −29.4 mV, −25.1 mV, and −26.7 mV, respectively. Although these values are just below the conventional ±30 mV threshold for high stability, they are substantial enough to provide significant electrostatic repulsion between particles, thus preventing rapid aggregation. The negative charge on these vesicles suggests the presence of anionic components on their surfaces, which is critical for maintaining their colloidal stability in an aqueous environment.

In contrast, the Standard CF Solution (SFAP250399) exhibited a zeta potential of −10.0 mV. This relatively low value indicates limited electrostatic repulsion, making the system more prone to aggregation and reduced long-term stability compared to the nanocarrier formulations.

In summary, the results demonstrate that the niosomal and liposomal formulations successfully encapsulate CF into stable, uniform nanovesicles, a feat not achieved by the standard solution, which exhibited a lower surface charge and reduced colloidal stability. Please refer to the Appendix A for further information on all samples.

### 3.4. Encapsulation Efficiency of Vesicle Formations:

The encapsulation efficiency (EE%) of the substance was calculated using the following equation:$$\mathrm{EE}\%\;=\;\frac{Amount\;of\;Encapsulated\;Substance}{Total\;Amount\;of\;Substance}\;\times \;100$$
where EE% represents the percentage of the total substance successfully encapsulated within the carrier system. Table 6 shows the results.

The synthetic niosome system exhibited the highest encapsulation efficiency at 43.65%, followed by the liposomal formation at 34.14%. The NN system showed the lowest efficiency, at 20.79%.

### 3.5. Biochemical Analysis for Physiological Assessment

Figure 2 demonstrates oxidative damage in different organs (A: kidney, B: heart, C: lung, D: liver).

Liver data sets were non-normally distributed and presented as median (min–max). These data sets were analyzed by using Dunn’s multiple comparison test as a post hoc analysis to find significant differences between groups after the Kruskal–Wallis test. Kidney, lung, and heart data sets were normally distributed and presented as mean ± SD. One-way ANOVA followed by Tukey’s test as a post hoc analysis was used for normally distributed sets.

Figure 3 shows inflammation levels in different organs (A: kidney, B: heart, C: lung, D: liver).

Kidney data sets were non-normally distributed and presented as median (min–max). These data sets were analyzed by using Dunn’s multiple comparison test as a post hoc analysis to find significant differences between groups after the Kruskal–Wallis test. Liver, lung, and heart data sets were normally distributed and presented as mean ± SD. One-way ANOVA followed by Tukey’s test as a post hoc analysis was used for normally distributed sets.

Figure 4 indicates levels of cellular damage (A: kidney, B: heart, C: lung, D: liver).

Heart data sets were non-normally distributed and presented as median (min–max). These data sets were analyzed by using Dunn’s multiple comparison test as a post hoc analysis to find significant differences between groups after the Kruskal–Wallis test. Liver, lung, and kidney data sets were normally distributed and presented as mean ± SD. One-way ANOVA followed by Tukey’s test as a post hoc analysis was used for normally distributed sets.

Figure 5 demonstrates organ dysfunction levels (A: kidney, B: heart, C: liver, D: lung).

Lactate, creatinine, ALT, and troponin data sets were normally distributed and presented as mean ± SD. One-way ANOVA followed by Tukey’s test as a post hoc analysis was used for normally distributed sets.

Figure 6 displays the kidney morphological status by micrographs and levels.

Normally distributed sets are presented as mean ± SD. One-way ANOVA followed by Tukey’s test as a post hoc analysis was used for normally distributed sets.

## 4. Discussion

The primary objective of this study was to evaluate the effectiveness of NN (natural niosome), a nanoparticle system developed entirely from natural processes, in delivering CF (calcium fructoborate) as the active compound. The findings provide several important insights into the structural stability, purity, and performance of this novel carrier system.

Elemental impurity analysis confirmed the purity of all tested formulations, including CF-loaded NN, synthetic niosome, liposome, and standard CF solutions. The absence of heavy metals such as Pb, Cd, and As, along with consistent levels of boron and calcium, underlines the reproducibility and elemental safety of these formulations. Such purity is essential for ensuring the biocompatibility and clinical relevance of nanocarrier systems.

The structural integrity of CF after encapsulation was also demonstrated. FT-IR spectra showed that characteristic functional groups were preserved across all nanocarrier types, with only minor spectral shifts attributable to interactions with lipid components. Together with particle size and zeta potential measurements, these findings highlight that CF was successfully encapsulated without compromising molecular identity. The nanocarriers exhibited nanometer-scale sizes, low PDIs (<0.25), and negative surface charges (−25 to −30 mV), all of which support stability, prevent aggregation, and ensure reliable biological performance. By contrast, the standard CF solution lacked vesicular structure, showed lower zeta potential, and was therefore less stable over time.

Despite these advantages, the NN formulation displayed the lowest encapsulation efficiency compared to synthetic niosomes and liposomes. However, this apparent limitation may actually translate into a more controlled and sustained release profile, which is often desirable in biomedical applications. Moreover, the natural origin of NN confers inherent benefits such as biocompatibility, biodegradability, and potentially reduced toxicity risks that are particularly important for long-term therapeutic use. In comparison, the higher encapsulation efficiency of synthetic niosomes raises questions regarding the long-term biological interactions of their artificial components, while liposomal systems, despite moderate efficiency, are associated with higher production costs and technical complexity. These comparisons emphasize the strategic value of developing natural-based nanocarrier systems.

Taken together, the results of this study suggest that while NN requires further optimization to improve encapsulation efficiency, its natural composition, safety profile, and potential for sustained release make it a promising candidate for future drug delivery applications.

In addition to these physicochemical characterizations, the study rigorously evaluated the anti-inflammatory and antioxidant properties of all drugs, alongside their potential for causing organ damage. The main finding of this study is that the NN formulation is associated with significant anti-inflammatory and antioxidant effects, particularly in abdominal organs, without causing any destructive impacts on them. These effects may not solely result from the nanoparticle delivery system itself but could also be partially attributed to the intrinsic bioactivities of natural components such as squalene and saponins used in the NN formulation. This robust and synergistic therapeutic efficacy, coupled with an excellent safety profile, is a major advantage of the naturally derived NN system. In stark contrast, while the commonly used liposome technique also exhibited anti-inflammatory properties, it was found to cause oxidative damage, especially in heart tissue. This highlights a critical safety concern of liposomal formulations compared to the benign nature of NN.

The ischemia–reperfusion model is a widely used experimental model for inducing acute inflammation [29]. In our study, ischemia was initiated by clamping the abdominal aorta, triggering oxidative stress and inflammation under hypoxic conditions. During this process, free radicals generated through anaerobic respiration due to oxygen deficiency may cause cellular peroxidation, and inflammation could be initiated via cytokines. In the ischemia–reperfusion model, oxidative stress continues to increase after the clamp is released during the reperfusion period [29]. This increase is mainly attributed to the imbalance in mitochondrial redox status during the anoxic phase [30].

In our study, CF exhibited antioxidant and anti-inflammatory effects by either preventing free radical formation or enhancing the activity of antioxidant enzymes [18]. However, the nanoparticle methods used altered CF’s efficacy during the processes of reaching and entering target cells. The dose of CF used in our study was determined based on the dose used in humans [17].

In our study, the antioxidant activity of different nanoparticles in the kidney, liver, heart, and lung tissues was compared using the MDA parameter. Since MDA is a lipid peroxidation product, an increase in its levels in tissues indicates oxidative stress damage. We observed that NN treatment reduced oxidative stress damage in the kidney, liver, and heart tissues better than all other methods. The primary reason for this is believed to be that the NN formulation, created in its natural form and containing molecules like saponins and squalene, transports the active ingredient more stably to the target cells, facilitating its easy uptake into the cells due to the nanoparticle’s natural form, which resembles the cell membrane.

Previous research has demonstrated that saponins and squalene have antioxidant and anti-inflammatory effects on their own [10,12]. Furthermore, the efficacy of the CF active ingredient has also been proven [30,31]. We believe that these results were achieved because the natural anti-inflammatory and antioxidant components in the structure of the natural form stabilized the active ingredient and facilitated its uptake into cells more effectively compared to other encapsulation methods.

Sialic acid, a glycocalyx component, has recently been used as an early-stage biomarker in oxidative stress studies [22]. In our study, similar to MDA levels, NN significantly prevented glycocalyx damage in the liver and kidneys, with the efficacy of the niosome being comparable to that of NN. However, liposome treatment was found to cause greater glycocalyx damage compared to the other groups. The absence of a corresponding difference in MDA levels for the liposome group is likely because sialic acid, as an early-stage marker, may be tolerated in the later stages. Moreover, we hypothesize that the early oxidative stress triggered by the liposome’s cholesterol content, potentially leading to rheological disturbances in capillaries, contributed to this effect. The study by Collin et al., published in 2020, supports this hypothesis regarding liposome effects [32]. Similarly, the increase in oxidative damage in heart tissue during liposome treatment aligns with this interpretation. According to the results, the damage increased to the point that it was higher than in the I/R group. Despite these results on cholesterol-containing liposomes, the cardiovascular protective effect of the NN form containing natural squalene, as reported in the study by Nurul et al., suggests that this formulation would likely show protective effects in this context [13].

The study by Romulus and Petre in 2011 reported that CF has anti-inflammatory effects on its own [17]. In our study, the evaluation of TNF-alpha levels measured in kidney, liver, lung, and heart tissues revealed that inflammation was suppressed in all organs by all methods. Contrary to the MDA results, TNF levels were most effectively suppressed in the liposome group in heart tissue. This discrepancy arises from the different pathways involved in inflammation and oxidative stress processes. As Forman and Zhang pointed out in their 2021 publication in Nature, oxidative stress does not always trigger inflammation, as inflammation is a cellular response that progresses through distinct pathways mediated by cytokines [33].

On the other hand, in abdominal organs such as the kidney and liver, NN was found to be more effective than the other treatments. We believe that this is due to the increased cellular permeability, which facilitates nanoparticle uptake into the cells. It is well known that niosomes enhance cell membrane permeability, allowing them to more effectively penetrate cells [34]. In our study, the kidneys, as representative abdominal organs, were subjected to ischemic injury via renal clamping. Therefore, endothelial permeability in these organs likely changed, allowing for NN to be taken up more effectively. This increased uptake, combined with the bioactive nature of NN’s natural components, may collectively contribute to its observed therapeutic advantages. As a result, the NN formulation was more effective in abdominal organs. Furthermore, the histological examination of the kidneys showed cellular morphological changes, including changes to epithelial thickness and tubular dilation, which Schelling indicated could affect permeability [35]. Although NN treatment appeared to reduce tubular dilation and epithelial thinning compared to the I/R group, these differences were not statistically significant. Therefore, while a trend toward histological improvement was observed, these findings should be interpreted with caution.

In our study, the short-term renal clamping method only led to organ dysfunction in the kidney tissue [36]. The evaluation of kidney function revealed that NN treatments, like all other treatment methods, were able to restore kidney function. The NN nanoparticle formulation, with its components such as squalene and saponin, proved to be therapeutic while also enhancing the ability of the active ingredient to reach the cells through the cell membrane [19,30]. CF, a substance with proven anti-inflammatory and antioxidant effects, was particularly effective in reversing dysfunction in kidney tissue in which ischemia–reperfusion damage was clearly observed when compared to other methods.

In summary, our study demonstrated that the NN nanoparticle method, produced entirely through natural processes using squalene and saponin, enhanced the efficacy of the CF active ingredient, although some histological improvements, such as reduced tubular dilation and epithelial thinning, did not reach statistical significance. These findings suggest a promising trend, but further studies with larger sample sizes are warranted to confirm these observations. Considering the oxidative stress damage caused by liposomal formulations in the heart, the importance of NN is clearly highlighted. In light of these results, NN, with its natural composition and low production cost, offers a promising alternative for use in treatments without causing damage to the organs.

## 5. Conclusions

This study rigorously compared different nanoparticle delivery methods and assessed their potential for organ damage, highlighting the significant advantages of the NN formulation. Produced through natural processes and at a remarkably lower cost, NN emerges as a promising new nanoparticle approach for drug delivery. Our findings demonstrate that NN has the potential to enhance therapeutic efficacy while mitigating adverse effects often associated with conventional methods. However, it is important to note that the observed benefits may not solely be attributed to the nanoparticle delivery method itself, but also to the intrinsic antioxidant and anti-inflammatory effects of its natural components, including squalene and saponins. This research paves the way for future investigations into novel, naturally derived nanoparticle drugs that are suitable for clinical application, offering a compelling and sustainable alternative in the field of drug delivery.

## Figures and Tables

**Figure 1 pharmaceutics-17-01434-f001:**
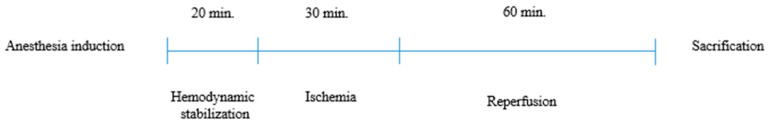
Surgical procedure.

**Figure 2 pharmaceutics-17-01434-f002:**
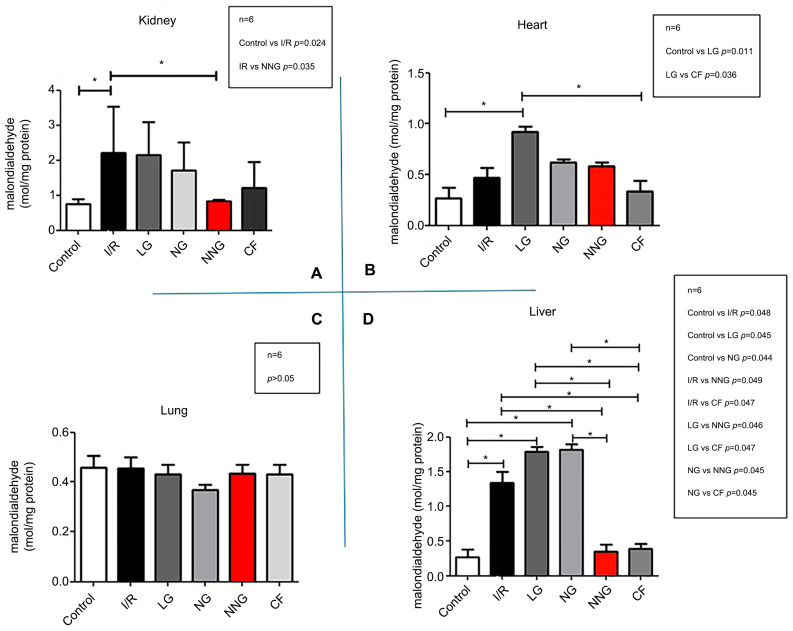
MDA levels were significantly higher in the I/R group (SG) compared with the control group (CG) and the natural niosome group (NNG) (* *p* < 0.05; control vs. I/R * *p* = 0.024, I/R vs. NNG * *p* = 0.035) in the kidney (**A**). MDA levels were significantly higher in LG than in CG and CF (* *p* < 0.05; control vs. LG * *p* = 0.011, LG vs. CF * *p* = 0.036) in the heart (**B**). There were no significant differences in MDA level between groups in the lung (*p* > 0.5) (**C**). In the liver, MDA levels were significantly higher in SG compared with CG, NNG, and CF, but LG and NG MDA levels were also higher compared with CG, NNG, and CF. (* *p* < 0.05; control vs. I/R * *p*= 0.048, control vs. LG * *p* = 0.045, control vs. NG * *p* = 0.044, I/R vs. NNG * *p* = 0.049, I/R vs. Cf * *p* = 0047, LG vs. NNG * *p* = 0.046, LG vs. Cf * *p* = 0.047, NG vs. NNG * *p* = 0.045, NG vs. CF * *p* = 0.045) (**D**).

**Figure 3 pharmaceutics-17-01434-f003:**
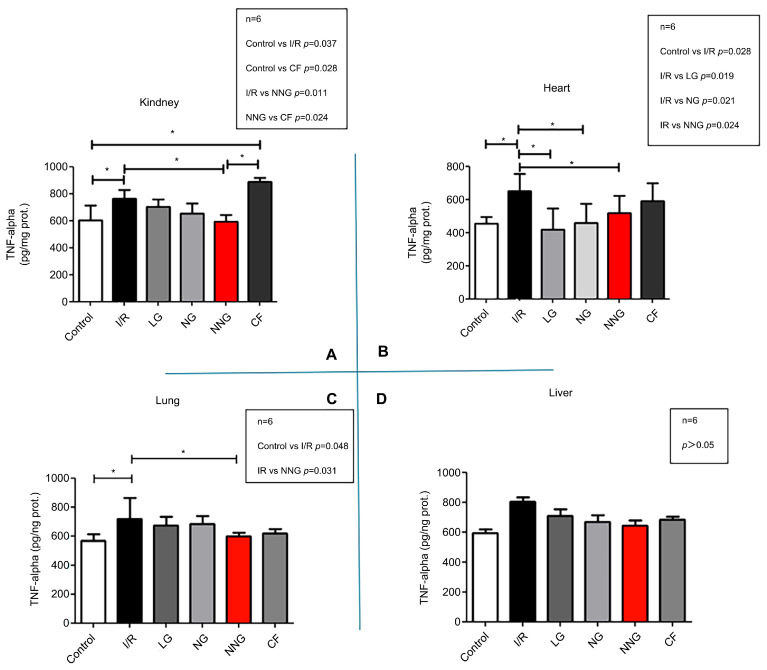
TNF-alpha levels were significantly higher in SG and CF than in CG and NNG in the kidney (control vs. I/R *p* = 0.037, control vs. CF * *p* = 0.028, I/R vs. NNG * *p* = 0.01, NNG vs. CF * *p* = 0.024) (**A**). In the heart, TNF-alpha levels were significantly higher in SG compared with CG, LP, NG, and NNG (* *p* < 0.05; control vs. I/R * *p* = 0.028, I/R vs. LG * *p* = 0.019, I/R vs. NG * *p* = 0.021, I/R vs. NNG * *p* = 0.024) (**B**). TNF-alpha was significantly higher in SG than in CG and NNG (*p* < 0.05; control vs. I/R* *p* = 0.048, I/R vs. NNG * *p* = 0.031) in the lung (**C**). There were no significant differences between groups for TNF-alpha levels in the liver (*p* > 0.05) (**D**).

**Figure 4 pharmaceutics-17-01434-f004:**
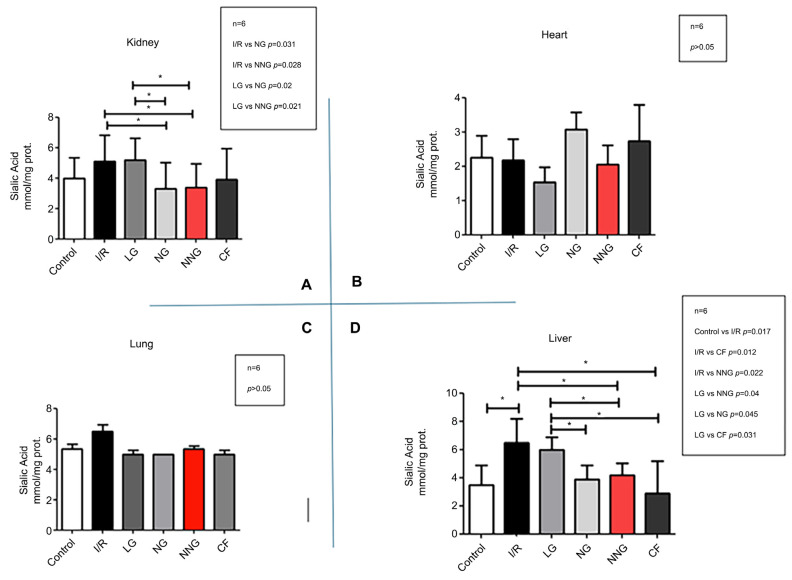
Sialic acid levels were significantly higher in SG and LG compared with NG and NNG (*p* < 0.05; I/R vs. NG *p* = 0.031, I/R vs. NNG *p* = 0.028, LG vs. NG *p* = 0.02, LG vs. NNG *p* = 0.021) in the kidney (**A**). There were no significant sialic acid level differences between groups in the heart and lung (*p* > 0.05) (**B**,**C**). In the liver, sialic acid levels were significantly higher in SG and LG compared with CG, NG, and NNG (* *p* < 0.05; control vs. I/R * *p* = 0.017, I/R vs. CF * *p* = 0.012, I/R vs. NNG * *p* = 0.022, LG vs. NNG * *p* = 0.04, LG vs. NG * *p* = 0.045, LG vs. CF * *p* = 0.031) (**D**).

**Figure 5 pharmaceutics-17-01434-f005:**
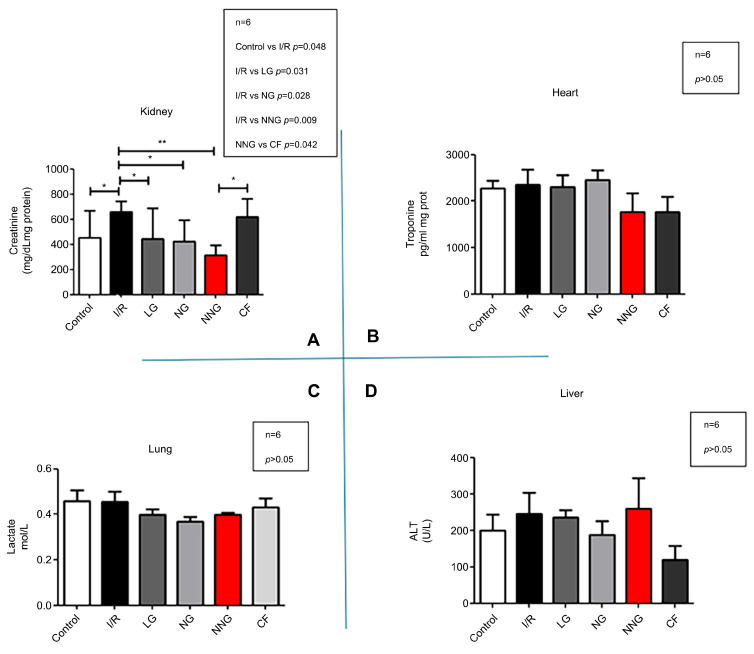
Creatinine levels were significantly higher in SG compared with LP, NG (*p* < 0.05), and NNG (*p* < 0.01). CF also had a significantly high value compared with NNG (*p* < 0.05; control vs. I/R * *p*= 0.048, I/R vs. LG * *p* = 0.031, I/R vs. NG * *p* = 0.028, I/R vs. NNG ** *p* = 0.009, NNG vs. CF * *p* = 0.042) in the kidney (**A**). There were no significant differences in ALT level between groups in the liver, as well as no significant differences in troponin level in the heart and lactate level in the lung (*p* > 0.05) (**B**–**D**).

**Figure 6 pharmaceutics-17-01434-f006:**
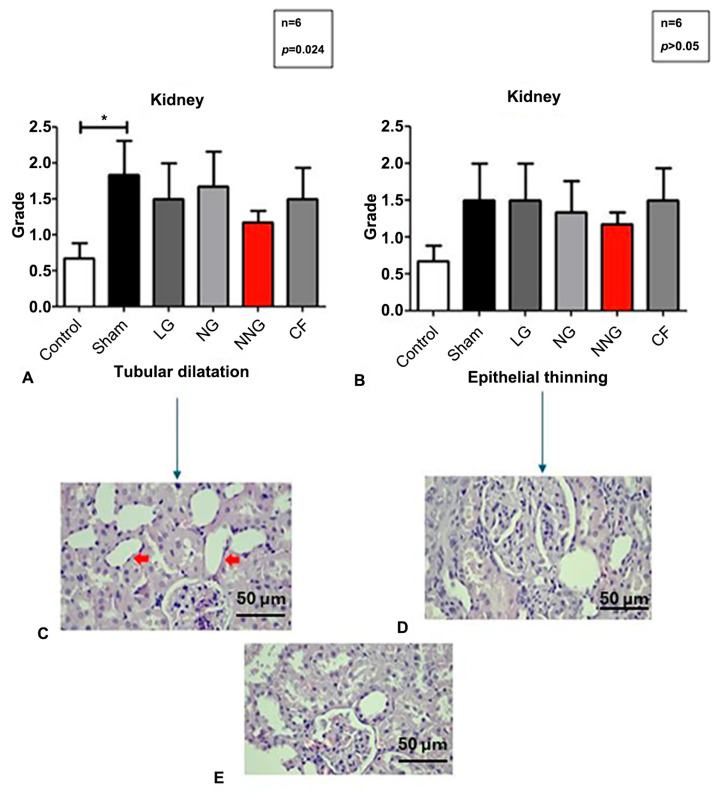
Although not statistically significant, tubular dilation levels were lower in the control and NNG, and epithelial thinning was higher in LG and lower in NNG (*p* > 0.05) (**A**,**B**). Microscopic images of tubular dilation in LG (H&E X 40) (**C**). Microscopic images of epithelial thinning in LG (H&E X 40) (**D**). Microscopic images of the kidney in NNG (H&E X 40) (**E**). Red arrows show the epithelial degradation in image (**C**).

**Table 1 pharmaceutics-17-01434-t001:** Concentrations of injection formulations and relevant stocks.

Formulation Type	Final Volume (mL)	CF Concentration (mg/mL)	Boron Concentration (mg/mL)	Notes
CF Stock Solution	—	3.0	0.077	Prepared from 40% (*w*/*w*) CF solution; used for dosing
Synthetic Niosomal CF	100	3.0	0.08 (target dose)	CF + Tween 80 + cholesterol (1:1:1); sonicated for 5 min
Natural Niosomal CF	100	3.0	0.08	CF + natural surfactant + cholesterol (1:1:1); similar to liposomes
Liposomal CF	100	3.0	0.08	CF + phospholipids (H90) + cholesterol (1:1:1); sonicated for 5 min
ICP-OES Standard Solution (CF Raw Material)	—	—	0.1617	Elemental boron content in pure CF raw material

**Table 2 pharmaceutics-17-01434-t002:** Experimental groups.

Group	Treatment	Ischemia	Reperfusion	Anesthesia
Control	-	-	-	Sodium pentothal (80 mg/kg i.p.)
I/R	-	30 min.	60 min.	Sodium pentothal (80 mg/kg i.p.)
LG	1 mL Liposomal CF (40%, *w*/*w*)	30 min.	60 min.	Sodium pentothal (80 mg/kg i.p.)
NG	1 mL Niosomal CF (40%, *w*/*w*)	30 min.	60 min.	Sodium pentothal (80 mg/kg i.p.)
NNG	1 mL Natural Niosomal CF (40%, *w*/*w*)	30 min.	60 min.	Sodium pentothal (80 mg/kg i.p.)
CF	1 mL CF (40%, *w*/*w*)	30 min.	60 min.	Sodium pentothal (80 mg/kg i.p.)

**Table 3 pharmaceutics-17-01434-t003:** Elemental composition of formulations and stocks (mg/L).

Sample	Natural Niosomal CF	Synthetic Niosomal CF	Liposomal CF	Standard CF
B	169.355	177.607	157.689	161.662
Ca	279.956	301.964	292.766	270.139
Fe	<0.010	<0.010	<0.010	<0.010
K	63.691	2.732	1.127	0.184
As	0.0089	0.0095	0.0131	0.0024
Be	0.0016	0.0027	0.0023	0.0013
Ba	0.0385	<0.001	<0.001	<0.001
Cr	0.0109	0.0149	0.0153	0.0062
Co	0.0030	0.0010	0.0028	<0.001
Cd	<0.001	<0.001	<0.001	<0.001
Cu	0.0598	0.0349	0.0209	0.0061
Pb	<0.001	0.0026	0.0014	0.0081
Mn	0.0639	0.0139	0.0169	0.0101
Ni	0.0047	<0.001	<0.001	0.0081
Zn	0.0838	0.0891	0.0820	0.0193

**Table 4 pharmaceutics-17-01434-t004:** Characteristic FT-IR peaks and corresponding functional groups (s: strong, w: weak, m: medium, vs: very strong). This table summarizes the main peaks observed in the FT-IR spectra of CF, along with their corresponding wavenumbers (in cm^−1^) and functional group assignments. This serves as a critical reference for understanding the molecular structure of the samples.

Spectrum	Wavenumber (cm^−1^)	Functional Group Assignment	Description
Reference CF from [28]	3521 s, 3401 s	O–H stretching, H_2_O stretching, C–H stretching	Broad bands indicating the presence of hydroxyl groups and water of crystallization.
Reference CF from Dumitru et al. (2010) [28]	2990 w, 2939 w	C–H stretching, CH_2_OH stretching	Peaks originating from the C–H bonds in the fructose skeleton.
All Spectra	~3256–3594	O–H stretching and H_2_O stretching	A broad, strong absorption band was observed in all samples, confirming the presence of hydroxyl groups and water within the molecule.
All Spectra	~2930–2990	C–H stretching and CH_2_OH stretching	Characteristic peaks for C–H bonds supporting the presence of the fructose molecule.
Reference CF from Dumitru et al. (2010) [28]	1652 w	B-O, H_2_O, OH, COH, OCH, CH_2_OH	A weak peak attributed to B–O bonds and other functional group vibrations.
Standard CF solution	1636.64, 1632.33	O–H bending, B–O stretching	Peaks that confirm the main compound, attributed to vibrations of water molecules and B–O bonds.
Liposomal CF	1683.97, 1653.85, 1646.68, 1636.64, 1625.16	O–H bending, stretching	The presence of multiple, more distinct peaks in this region compared to the reference spectrum may indicate interactions related to liposomal encapsulation.
Synthetic Niosomal CF	1683.97, 1679.67, 1675.36, 1668.19, 1653.85, 1646.68, 1636.64, 1629.46, 1625.16	O–H bending, B–O stretching	Similar to the liposomal form, the multiplicity of peaks in this region suggests the influence of niosomal encapsulation on molecular vibrations.
Natural Niosomal CF	1683.97, 1679.67, 1652.41, 1646.68, 1636.64	O–H bending, B–O stretching	This sample shows similar peaks to the niosomal form, reflecting the interactions of molecular vibrations with the encapsulation process.

**Table 5 pharmaceutics-17-01434-t005:** Summary of particle size and zeta potential characterization results.

Sample Name	Z-Average (nm)	PDI	Mean Zeta Potential (mV)
Natural Niosomal CF (SFAP250394)	152.0	0.168	−29.4
Liposomal CF (SFAP250395)	88.29	0.248	−25.1
Synthetic Niosomal CF (SFAP250397)	136.8	0.175	−26.7
Standard CF Solution (SFAP250399) (Control)	111.7	0.602	−10.0

**Table 6 pharmaceutics-17-01434-t006:** Encapsulation efficiency (EE%) of niosome and liposome systems.

Vesicle	Encapsulated Amount (Mean ± SD) mg	Total Amount (mg)	EE%
**Natural Niosome**	4867 ± 1.17	23,405	20.79%
**Synthetic Niosome**	10,493 ± 1.21	24,043	43.65%
**Liposome**	5685 ± 0.25	16,654	34.14%

## Data Availability

The original contributions presented in this study are included in the article/Appendix A. Further inquiries can be directed to the corresponding author(s).

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
