# Peer review of "Efficiency of Calcium Fructoborate-Loaded Novel Natural Niosomes Compared to Traditional Liposomes and Niosomes in Rat Ischemia–Reperfusion Injury Model"

_pharmaceutics, 2025, doi:10.3390/pharmaceutics17111434_

Round 1
Reviewer 1 Report
Comments and Suggestions for Authors
The main objective of this study was to evaluate the effectiveness of novel natural niosome (NN) system in comparison to existing liposomal nanocarriers formulations for an ischemia-reperfusion pain model. The research presented in the paper is interesting, but the description of the physicochemical methods and the physicochemical properties of the nanocarriers is insufficient.
Please standardise the abbreviations/descriptions of the nanocarriers used in different parts of the manuscript, e.g. Table 1 and Table 2. Similarly, the description in the introduction and methodology.
Please provide information on how long after the formation of the nanocarriers the measurements of the polydispersity coefficient and zeta potential were performed. What is the stability of the newly obtained formulations?
What is the boron encapsulation efficiency in individual formulations?
The authors state that "FTIR spectroscopy was performed to investigate potential interactions of liposomal 200 CF and the CF molecule". The FTIR spectra presented have not been described in detail. Please analyse all visible signals. Were measurements performed for NN? Please include a complete set of results with the values of individual IR signals marked and specified.
Author Response
On behalf of all co-authors, I would like to thank you and the reviewers for their constructive and valuable comments on our manuscript. We carefully revised the paper and addressed each of the points raised. Below, I summarize the main revisions. All revisions are highlighted in yellow in the revised manuscript.
Reviewer 1:
We repeated the zeta potential and polydispersity coefficient measurements and included the updated data in the revised version. Detailed results are now provided in the Supporting Information.
Encapsulation efficiency of boron was experimentally determined. A new subsection describing the methodology and results has been added to the Methods and Results sections, and the outcomes are discussed in the Discussion.
The FTIR data were renewed. All visible signals have been described in detail, and the specific spectra have been marked, analyzed, and included in the Supporting Information.

Reviewer 2 Report
Comments and Suggestions for Authors
The manuscript addresses an important area in nanomedicine: the development of a novel natural niosome (NN) formulation using squalene and saponins as stabilizers for calcium fructoborate (CF), tested in a rat ischemia-reperfusion (I/R) injury model.
There are several comments:
-
- A vehicle-only natural niosome (without CF) control is missing. This is critical since squalene and saponins themselves may have biological activity.
- In Table 2, the control group is listed as "no ischemia, no reperfusion, no treatment under sodium pentothal anesthesia. Sham group is described as “ischemia + reperfusion, but without drug treatment". In general, the Sham group is an animal that undergoes the mimic surgical procedure without inducing ischemia or reperfusion. So this experiment lacks a real sham group. Please rename your groups for accuracy and add a real sham group.
-
- The text sometimes confuses correlation with causation (e.g., attributing all protective effects directly to NN without acknowledging possible inherent effects of saponins/squalene).
- Statistical reporting should be improved (exact p-values rather than only thresholds, clearer indication of tests used for each dataset).
- The manuscript states NN reduced tubular dilation and epithelial thinning, but the differences were not statistically significant. This should be clearly acknowledged in the discussion to avoid overstatement.
- In the text, it is distributed data were presented as mean ± SD, while others were presented as median (min–max). However, this distinction is not clearly indicated in the figure panels or legends. It is difficult for readers to understand. To improve transparency, please clearly label in each figure legend whether the data are expressed as mean ± SD or median (min–max). If possible, provide exact n values on the graphs to indicate sample size.
- The supplementary data show particle size, PdI, and zeta potential for liposomes, Tween-80 niosomes, and natural niosomes. These results support the higher stability of NN. Still, the following issues arise such as PdI of liposomes is relatively high (>0.5), suggesting instability. Was this preparation method optimized?
Overall, English is acceptable, but should be polished for clearity and conciseness.
Author Response
On behalf of all co-authors, I would like to thank you and the reviewers for their constructive and valuable comments on our manuscript. We carefully revised the paper and addressed each of the points raised. Below, I summarize the main revisions. All revisions are highlighted in yellow in the revised manuscript.
We would like to express our gratitude for pointing out the issue with the Sham group. Due to a technical error during the writing process, the groups labeled as "Sham" were actually part of the Ischemia/Reperfusion (I/R) injury group. This correction has been made throughout the manuscript. As a result, as outlined in Table 2, the groups in our study are Control, I/R, LP, NNG, NG, and CF.
In designing the study, healthy, diseased, and treatment groups were planned in accordance with the study's objectives, and a Sham group was not included. The Sham group is typically used to isolate the effects of the surgical procedure itself, but since the I/R group already involves the surgical procedure along with ischemia and reperfusion, the addition of a Sham group does not provide meaningful information for our study.
Furthermore, the focus of our study is not to elucidate the pathophysiological mechanisms, but rather to investigate the improvement of inflammation induced by the I/R process. Therefore, evaluations made using the I/R group are sufficient for the objectives of our study.
Biochemical analyses show that the I/R group developed the pain model and triggered inflammation. The healthy control group was used to observe the resolution of this pain and inflammation model and to evaluate whether treatment had been effective.
In conclusion, a Sham group is not necessary, as the I/R group already provides the required comparisons for both the surgical effects and the inflammation and healing processes.
Since CF is kept constant across all treatment groups, we observed the results of the therapeutic effects of the natural niosome or its potential to enhance drug efficacy through the comparisons we made in the LP and NG groups. We agree that saponins and squalene may have anti-inflammatory effects on their own, and for this reason, these components were selected for the formulation of the natural niosome. However, in our study, rather than evaluating CF and NNG structures separately, the aim was to compare the synergistic anti-inflammatory activities of these compounds together with those of nanoparticles produced by traditional methods. Therefore, by considering the potential effects of CF and NNG as a whole, we aimed to measure the combined efficacy of these two compounds in a more meaningful way.
We sincerely thank the reviewer for comments 3, 4, 5, and 6. In response, the manuscript has been carefully revised: the relevant sections have been modified to clearly distinguish correlation from causation and to acknowledge the possible inherent effects of saponins and squalene; exact p-values and the statistical tests used for each dataset have been provided; the Discussion section has been updated to state explicitly when differences did not reach statistical significance; and all figure legends now clearly indicate whether data are expressed as mean ± SD or median (min–max), with exact sample sizes (n) added to the graphs for transparency. All these changes have been incorporated into the revised version and highlighted in yellow.
The zeta potential values were re-measured, and the updated data are included in the Supporting Information. Upon re-evaluation, we agreed with the reviewer’s concern regarding the relatively high PdI values. The initial data indicated that the formulation was not optimized. However, in the renewed measurements, the PdI values were found to be within normal ranges, which has now been reflected in the revised manuscript and supporting data.
We apologize for the delay in resubmission. The revision process took longer due to the heavy workload and scheduling constraints in the university laboratories.
We believe that the revised version addresses all reviewer concerns and significantly strengthens the manuscript. We sincerely appreciate the reviewers’ insightful feedback and the editorial support provided.
Thank you for your consideration.
